# CooccurrenceAffinity: An R package for computing a novel metric of affinity in co-occurrence data that corrects for pervasive errors in traditional indices

Kumar P. Mainali[1,2]*, Eric Slud[3,4]

**1** Conservation Innovation Center, Chesapeake Conservancy, Earl Conservation Center, Annapolis, Maryland, United States of America, **2** Department of Biology, University of Maryland, Annapolis, Maryland, United States of America, **3** Department of Mathematics, University of Maryland, College Park, Maryland, United States of America, **4** Center for Statistical Research and Methodology, US Census Bureau, Washington, DC, United States of America

\* kpmainali@utexas.edu

**Data Availability Statement:** All relevant data are within the paper.

**Funding:** KM was supported by the Grayce B. Kerr Fund, Inc, and by the National Science Foundation

## Abstract

1. Analysis of co-occurrence data with traditional indices has led to many problems such as sensitivity of the indices to prevalence and the same value representing either a strong positive or strong negative association across different datasets. In our recent study (Mainali et al 2022), we revealed the source of the problems that make the traditional indices fundamentally flawed and unreliable–namely that the indices in common use have no target of estimation quantifying degree of association in the non-null case–and we further developed a novel parameter of association, alpha, with complete formulation of the null distribution for estimating the mechanism of affinity. We also developed the maximum likelihood estimate (MLE) of alpha in our previous study.

2. Here, we introduce the CooccurrenceAffinity R package that computes the MLE for alpha. We provide functions to perform the analysis based on a 2×2 contingency table of occurrence/co-occurrence counts as well as a m×n presence-absence matrix (e.g., species by site matrix). The flexibility of the function allows a user to compute the alpha MLE for entity pairs on matrix columns based on presence-absence states recorded in the matrix rows, or for entity pairs on matrix rows based on presence-absence recorded in columns. We also provide functions for plotting the computed indices.

3. As novel components of this software paper not reported in the original study, we present theoretical discussion of a median interval and of four types of confidence intervals. We further develop functions (a) to compute those intervals, (b) to evaluate their true coverage probability of enclosing the population parameter, and (c) to generate figures.

4. CooccurrenceAffinity is a practical and efficient R package with user-friendly functions for end-to-end analysis and plotting of co-occurrence data in various formats, making it possible to compute the recently developed metric of alpha MLE as well as its median and confidence intervals introduced in this paper. The package supplements its main output of the

DBI-1639145 under funding received for the
National Socio-Environmental Synthesis Center
(SESYNC). The funders had no role in study
design, data collection and analysis, decision to
publish, or preparation of the manuscript.

**Competing interests:** The authors have declared
that no competing interests exist.

novel metric of association with the three most common traditional indices of association in
co-occurrence data: Jaccard, Sørensen–Dice, and Simpson.

## Introduction

Co-occurrence data are analyzed to estimate similarities among entities in many disciplines
including biogeography [1,2], biodiversity [3,4], ecology [5], epidemiology [6], evolution [7],
and neuroscience [8]. Over a century's quantitative development has produced about 80 met-
rics of association in co-occurrence data [9], of which the most popular metrics are Jaccard,
Sørensen–Dice, and Simpson [10]. In a recent study [11], we showed that these indices of asso-
ciation suffer from fundamental statistical flaws, making them unreliable as a measure of asso-
ciation. These flaws (manifesting differently for different indices) are two-fold: that the indices
depend sensitively on prevalence and fail to group similar non-null associations together. For
example, we show situations when the same value of Jaccard's Index (e.g., 0.6) indicates strong
negative and strong positive association (see Fig 1D in [11]). Many publications about beta
diversity have reflected this lack of reliability of the traditional indices [12]. See [11] for a
detailed account of problems of traditional indices from the viewpoint of statistical theory as
well as of ecological practice.

We recently resolved the challenges of traditional indices by developing a reliable, meaning-
ful and interpretable parameter of association in binary co-occurrence data. We developed this
novel parameter, named alpha ($\alpha$), from solid statistical theory and presented a complete
mathematical formulation of its probability distribution in [11]. The purpose of this software
is fourfold: (1) to compute and display a point estimate (Maximum Likelihood Estimate) of $\alpha$
or *log-affinity* as advanced in [11], (2) to compute and present the median interval and four
types of confidence intervals, (3) to develop a pipeline to analyze data that comes either in a
standard 2×2 contingency table of co-occurrences and occurrences or in a matrix of presence-
absence of entities by elements (e.g., species by site matrix), and (4) to generate plots with easy
customization of graphical elements.

## Models, parameters and intervals

Imagine an ecological example involving distribution of two species on a grid of $N$ sites in a
specified locale. The occurrence data for each species pair typically constitute a 2×2 table with
fixed margins (Fig 1a and 1b) where $m_A$, $m_B$ are the respective numbers of ecological sites
occupied by species A and B (prevalence of Species A = $m_A/N$). The figure illustrates two types
of notations used to represent the number of sites that contain both species, only species A,
only species B and neither of them.

The probability model for these data is a 'balls-in-boxes' or Urn Model. It encompasses
both the null-hypothetical model of random assortment subject to given prevalences $m_A$, $m_B$
and numbers of sites ($N$), as well as a single-parameter model expressing a quantitative affinity
relating to the probabilities with which the two species occupy each of the sites.

### The parameter

The model assumes that species A occupies $m_A$ sites out of the $N$ equiprobably, meaning each
set of $m_A$ sites is equally likely to be occupied. Species B then occupies sites by choosing them
independently of one another, with probability $p_1$ if the site is already occupied by species A
and with probability $p_2$ if the site is not occupied (Fig 2a). The random variable $X$ of interest

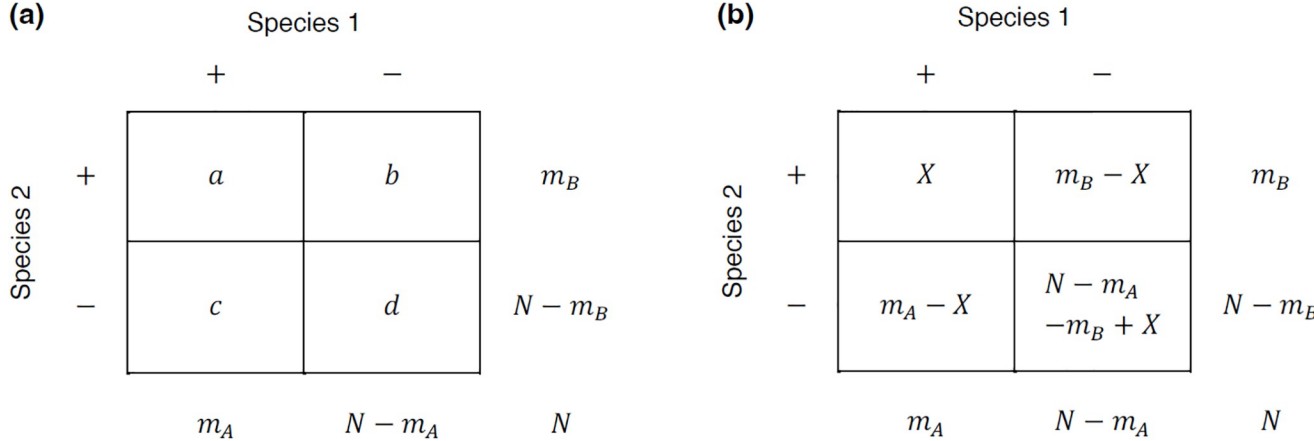

**Fig 1. Two ways of displaying a 2×2 table of co-occurrences and occurrences of two entities with fixed margins.** The example illustrates the distribution of Species A and B occupying $m_A$ and $m_B$ sites, respectively, out of a total $N$ sites. The number of sites where both species co-occur (top-left quadrant), where only Species A exists (bottom-left quadrant), where only Species B exists (top-right quadrant), and where none of the species exist (bottom-right quadrant) typically appear in popular scientific literature with one set of notations (Fig 1a) but often appear in statistical literature with different notations (Fig 1b).

represents the number of sites occupied by both species A and B. The probability distribution of $X$ is computed conditionally, given that the total number of sites occupied by species B is fixed at $m_B$. This probability distribution is known as the Extended Hypergeometric Distribution [13] or Noncentral Hypergeometric Distribution [14]. It turns out that this distribution would be the same if the roles of A, B were reversed (i.e., the $m_B$ sites occupied by B were picked first). The distribution depends on the probabilities $p_1$, $p_2$ only through the single log odds-ratio parameter:

$$\alpha = \log\left(\frac{p_1(1 - p_2)}{p_2(1 - p_1)}\right), \tag{1}$$

and has the following probability mass function (conditional on $m_A$, $m_B$, $N$) for $\max(m_A + m_B - N, 0) \leq k \leq \min(m_A, m_B)$:

$$P(X = k) = \frac{\binom{m_A}{k}\binom{N - m_A}{m_B - k}e^{k\alpha}}{\sum_{j=\max(m_A+m_B-N,0)}^{\min(m_A,m_B)}\binom{m_A}{j}\binom{N - m_A}{m_B - j}e^{j\alpha}}. \tag{2}$$

The parameter $\alpha$ defined in Eq (1) is called log odds-ratio in standard statistical presentations of categorical data, and is ¼ times the row-column interaction parameter in standard parameterizations of *loglinear models* [15] on a 2×2 table. In the ecological context, Mainali et al. (2022) recently advanced this unknown statistical parameter to quantify the extent to which species A, B appear together or to quantify the affinity between any two entities based on their co-occurrence pattern. The statistical estimate of this parameter by maximum likelihood is called alpha hat ($\hat{\alpha}$) in [11] and is shown there to be much preferable to standard ecological indices of species-pair association or beta diversity because it quantifies dependence in a way insensitive to and interpretable apart from prevalences.

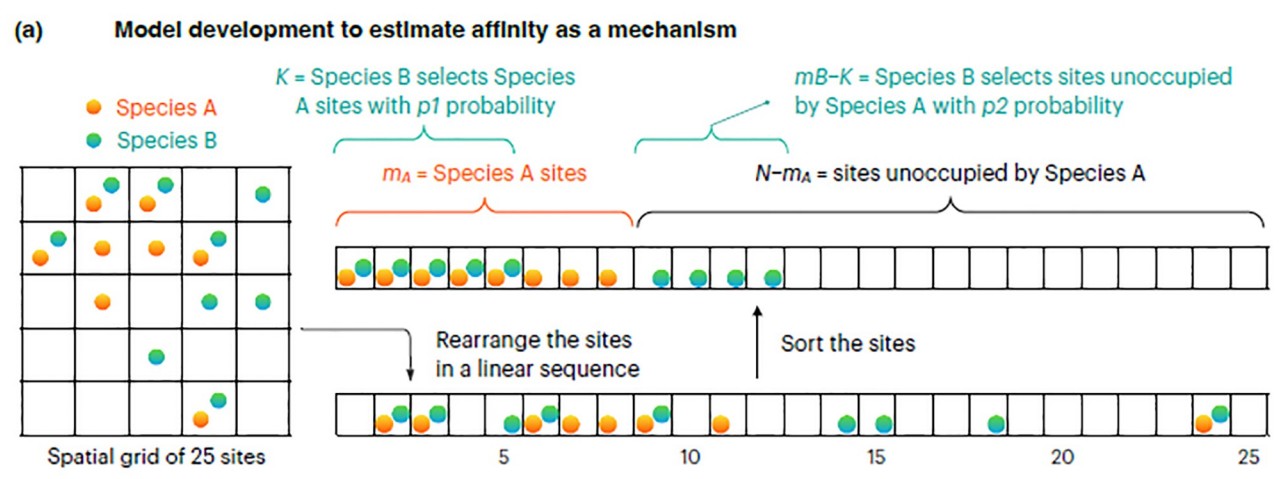

## (a) Model development to estimate affinity as a mechanism

- Species A
- Species B

K = Species B selects Species A sites with p1 probability

mB−K = Species B selects sites unoccupied by Species A with p2 probability

$m_A$ = Species A sites

$N-m_A$ = sites unoccupied by Species A

Spatial grid of 25 sites

Rearrange the sites in a linear sequence

Sort the sites

## (b) When two species occur Independent of each other: Balls-In-boxes analogy and null model

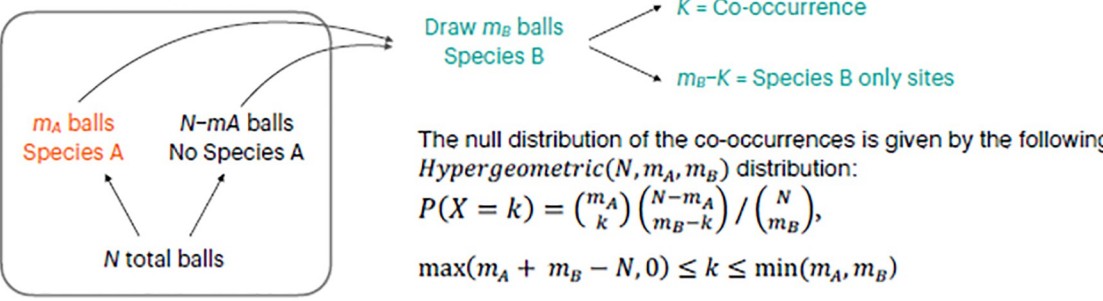

Draw $m_B$ balls Species B

K = Co-occurrence

$m_B$−K = Species B only sites

$m_A$ balls Species A

$N-m_A$ balls No Species A

$N$ total balls

The null distribution of the co-occurrences is given by the following $Hypergeometric(N, m_A, m_B)$ distribution:

$$P(X = k) = \binom{m_A}{k}\binom{N-m_A}{m_B-k} / \binom{N}{m_B},$$

$$\max(m_A + m_B - N, 0) \le k \le \min(m_A, m_B)$$

## (c) When two species have a positive or a negative affinity

The degree of difference of the two probabilities through the following log odds ratio quantifies the affinity of the two species to each other:

$$\alpha = log\left(\frac{p_1}{1 - p_1} \middle/ \frac{p_2}{1 - p_2}\right)$$

The probability of exactly $X = k$ and $N_B = m_B$ under affinity is

$$\binom{m_A}{k} \cdot p_1^k (1 - p_1)^{m_A - k} \cdot \binom{N-m_A}{m_B - k} p_2^{m_B - k} (1 - p_2)^{N - m_A - m_B + k}$$

The probability of co-occurrence under affinity depends on $m_A$, $m_B$, $N$ and $\alpha$, but not on $p_1$ or $p_2$, having the more general form:

$$P(X = k \mid N_B = m_B) = \binom{m_A}{k}\binom{N-m_A}{m_B-k} e^{\alpha k} / \sum_{j=0}^{m_B} \binom{m_A}{j}\binom{N-m_A}{m_B-j} e^{\alpha j},$$

$$\max(m_A + m_B - N, 0) \le k \le \min(m_A, m_B).$$

This is the Extended Hypergeometric or noncentral hypergeometric distribution. A complete formulation of these solutions, as well as Maximum Likelihood Estimator of $\alpha$ has been reported in Mainali et al (2022).

**Fig 2. Model development for affinity between two species.** The three stages illustrate how the notion of affinity was formulated mathematically (a). The null model for co-occurrence counts in the situation of zero affinity emerges as the Hypergeometric distribution (b) and extends in the case of nonzero affinity to the Extended Hypergeometric distribution, leading to an estimable parameter $\alpha$ that governs association in the 2×2 table.

The complete mathematical formulation of $\hat{\alpha}$ metric is available in [11]. In Fig 2, we present a semi-graphical summary of the development of the $\hat{\alpha}$ metric. For the scenario of two species described above, the log odds ratio—the log of the odds that Species B appears in sites occupied by species A relative to sites not occupied by species A—is the most intuitive and mechanistic expression of the true affinity between the two species (Fig 2a). When the two species have zero affinity to each other, the arrangement in Fig 2a naturally aligns with the "balls-in-boxes" description of the Hypergeometric null distribution (Fig 2b). The mathematical form of the null distribution appeared first within ecological literature in papers of [16,17]. When the two species have some affinity (positive or negative) to each other, their selective preference modifies the Hypergeometric distribution as shown in Fig 2c, to yield the Extended Hypergeometric distribution (Fig 2c).

## The maximum likelihood estimator

The Maximum Likelihood Estimator (MLE), $\hat{\alpha}$, is a function of the data ($X, m_A, m_B, N$), computed by numerically maximizing the right-hand side of Eq (2) with $k$ replaced by the observed $X$ value. It has all the usual favorable properties of maximum likelihood estimators, including an approximate normal distribution when the sample sizes are large. In this context, "large" means that $\min(m_A, m_B, N - m_A, N - m_B)$ is sufficiently large. A property of MLE is that it can be slightly biased in small- and moderate-sized data. Because MLEs are typically used to define similarity of entity-pairs with respect to degree of association, they are meaningful even when samples are not large.

## Confidence intervals and coverage probability

Along with the MLE, the software computes confidence intervals for the unknown parameter. Confidence intervals, which take the form $(-\infty, a)$ or $(a, b)$ or $(b, \infty)$ with endpoint(s) determined from the data, are intended to bracket the true population parameter value. These intervals are specified along with a nominal coverage probability (usually 95%) quantifying the desired probability that the interval will contain the unknown population parameter ($\alpha$, in our case).

The actual probability that an interval, based on data satisfying the assumed model, encloses the true population parameter is called the true coverage probability. In discrete data settings, the true coverage probability cannot be guaranteed to be equal to the nominal probability. The investigator may prefer to maintain the true and nominal coverage probability to be as close as possible, or may feel bound by the constraint that true coverage probability should always be at least as large as the nominal. But one cannot generally achieve both.

## Special handling of the most extreme co-occurrences

When $k = \max(m_A + m_B - N, 0)$ or $k = \min(m_A, m_B)$, the extended-real-valued maximizer of Eq (2) occurs, respectively, at $\alpha = -\infty$ or $\alpha = +\infty$. That is, the extreme co-occurrence count $k$ is compatible with arbitrarily large log odds ratio, respectively $-\infty$ or $+\infty$. This is a feature of extremely inconclusive data, not a faulty method of estimation. To avoid these infinite values, we follow a Bayesian argument sketched in equation 8 of [11] and modify the definition of the MLE $\hat{\alpha}$ in the case where $X$ is one of these extreme endpoints respectively to $\mp\log(2N^2)$. When reporting such results, it is important to exercise caution, particularly when plotting capped values $\mp\log(2N^2)$ alongside other values. The trend observed in the uncapped values may not necessarily extend to the large finite capped values, which should be viewed as essentially infinite. It is crucial to avoid interpreting this discontinuity as indicative of the system under

study. Instead, it should be recognized as an artifact resulting from how our software handles special cases by providing a more useful value rather than an undefined $-\infty$ or $+\infty$.

For aggregated analysis involving both capped and uncapped values, we recommend reporting the finite endpoints of one-sided 95% confidence intervals, because of the unboundedly large values of $\alpha$ compatible with the data. Specifically, a 95% interval of the form $(b, \infty)$ should be reported when the co-occurrence count is at its highest extreme, and a 95% interval of the form $(-\infty, a)$ should be reported when the co-occurrence count is at its lowest extreme. This approach results in a quantity (finite endpoint of the CI) that does not suffer from truncation, making it reliable for comparison across the dataset containing cases with the most extreme co-occurrence values allowed by their respective species prevalences as well as other cases.

## Advances: The median interval

When sample sizes are small the discreteness of the probability distribution (2) is an obstacle to precise estimation of $\alpha$ or confidence intervals covering it. One aspect of this discreteness is that for each q $\epsilon$ (0, 1) and each $k$ satisfying $\max(m_A + m_B - N, 0) < k < \min(m_A, m_B)$, there is not a single value but rather an interval of values for $\alpha$ that are compatible with $k$ being a $q$'th quantile of (2), defined by the simultaneous inequalities

$$P(\mathrm{X} \leq x \mid m_A, m_B, N, \alpha) \geq \mathrm{q}, P(\mathrm{X} \geq x \mid m_A, m_B, N, \alpha) \geq 1 - \mathrm{q} \qquad (3)$$

Denote by $F(x, \alpha) = P(\mathrm{X} \leq x \mid m_A, m_B, N, \alpha) = F(x, m_A, m_B, N, \alpha)$ the Extended Hypergeometric distribution function for the random variable $X$ which is the upper-left entry of a 2×2 table with fixed first row-sum $m_A$, first column-sum $m_B$, table-total $N$, and odds-ratio parameter $e^\alpha$. It is easy to verify from (2) that $F(x, \alpha)$ is strictly monotonically decreasing in $\alpha$, converging to 0 when $\alpha \longrightarrow \infty$ and to 1 when $\alpha \longrightarrow -\infty$ for $\max(m_A + m_B - N, 0) < x < \min(m_A, m_B)$. For such $x$, we denote by $F(x, \cdot)^{-1}$ the inverse function with respect to $\alpha$, from which it follows immediately that the largest open interval of $\alpha$ values satisfying (4) is $(F(x - 1, \cdot)^{-1}(p), F(x, \cdot)^{-1}(p))$, where $F(z, \cdot)^{-1}(p)$ is defined equal to that value $a$ for which $F(z, a) = p$. All quantities $F(x, a)$ computed in the CooccurrenceAffinity package are obtained from the function pFNCHypergeo in the R package BiasedUrn. Note that, for reasons of numerical accuracy and convergence of the functions in the BiasedUrn package, all absolute values of $\hat{\alpha}$ are capped at the largest value less than or equal to 10 for which the BiasedUrn functions work, usually 10 or close to it.

The interval described in the previous paragraph for $x = X$ and $p = 1/2$ will be called the *median interval* for $\alpha$ based on $X$. This interval in our experience always contains the MLE of $\hat{\alpha}$ (an empirical finding, always found to hold numerically but not yet proved mathematically), and its length quantifies the discreteness in the Extended Hypergeometric distribution and thus the ambiguity of $\hat{\alpha}$.

## Advances: Four types of confidence intervals

Next, we discuss the computation of the (two-sided equal-tailed) Confidence Interval for $\alpha$ based on the data $(X, m_A, m_B, N)$. The concept of "test-based confidence interval" is central to explaining the confidence intervals for log-affinity parameter computed in this package. This idea is explained well in [[18], Sec.9.2.1] under the heading "Inverting a Test" and is familiar– sometimes under different names–from the history of confidence intervals for an unknown binomial proportion. Essentially the same idea is standard in defining so-called 'exact' confidence intervals for an unknown scalar parameter that is monotonically related to the distribution function for a discrete random variable. This construction applies to the parameter for

Binomial, Poisson, Geometric, and Negative Binomial random variables, as well as for the log odds-ratio or affinity parameter $\alpha$ of the Extended Hypergeometric [13] random variable, but we describe it in detail only in the affinity case.

The common feature shared by all of these discrete (integer-valued) random variables $X$ with parameter $\theta$ is the Monotone Likelihood Ratio (MLR) property [[18], p.391], which states that the ratio $p(x, \theta_1) / p(x, \theta_2)$ for fixed $\theta_1 < \theta_2$ is a monotonic function of the integers $x$. This property, which is also a consequence of the natural-exponential-family property shared by these random variables, implies that the rejection regions for optimal (Neyman-Pearson) hypothesis tests of $H_0: \theta \leq \theta_0$ versus $H_1: \theta > \theta_0$ or of $H'_0 : \theta \geq \theta_0$ versus $H'_1 : \theta < \theta_0$ are respectively intervals $\{x: x \geq k_2\}$ or $\{x: x \leq k_1\}$. This is (part of) the Karlin-Rubin Theorem [[18], Theorem 8.3.17]. In view of this discussion along with the monotone decrease of $F(x, \alpha)$ with respect to $\alpha$, a natural equal-tailed test with significance level $\gamma$ for the point null-hypothesis $H_0: \alpha = \alpha_0$ versus the two-sided alternative $H_A: \alpha \neq \alpha_0$, would reject when $X$ is too large or small, i.e., when $X \leq k_1$ or $X \geq k_2$ where the cut-offs $k_1, k_2$ are respectively determined as

$$k_1 = \max\left\{k : F(k, \alpha_0) \leq \frac{\gamma}{2}\right\}, \; k_2 = \min\{k + 1 : F(k, \alpha_0) \geq 1 - \gamma/2\}$$

Thus, the acceptance region (complement of rejection region) for the test is the set of x's for which

$$F(x, \alpha_0) > \gamma/2 \text{ and } F(x - 1, \alpha_0) < 1 - \gamma/2.$$

**Clopper-Pearson type confidence interval (CI.CP).** In terms of this family of hypothesis tests, the idea of the "test-based" or "inverted" confidence intervals for $\alpha$ based on $X$ is to take as confidence interval $CI(X, 1 - \gamma)$ of confidence-level $1 - \gamma$ the set of parameter values $\alpha_0$ for which the corresponding hypothesis tests would accept $H_0: \alpha = \alpha_0$. The idea of the test-based interval is that the Confidence Interval is the set of $\alpha_0$ values compatible with the data $X$.

This confidence set is an interval because $F(x, \alpha)$ is strictly decreasing with respect to $\alpha$. Suppose that $x$ is a possible value of the Extended Hypergeometric random variable, which is equivalent to saying $\max(m_A + m_B - N, 0) \leq x \leq \min(m_A, m_B)$. Recall that $F(x, \cdot)^{-1}(r)$ for $0 < r < 1$ is the unique value of $\alpha \epsilon (-\infty, \infty)$ for which $F(x, \alpha) = F(x, m_A, m_B, N, \alpha) = r$. Then the acceptance region for the test $H_0: \alpha = \alpha_0$ above is the same as

$$\{x : F(x - 1, \cdot)^{-1}(1 - \gamma/2) < \alpha_0 < F(x, \cdot)^{-1}(\gamma/2)\}$$

or as the *Clopper-Pearson type Confidence Interval (CI.CP)*

$$\alpha_0 \epsilon CI(X, 1 - \gamma) = (F(X - 1, \cdot)^{-1}(1 - \gamma/2), F(X, \cdot)^{-1}(\gamma/2)) \tag{4}$$

The statement that the test $H_0: \alpha = \alpha_0$ has significance level $\gamma$ is exactly the same as the statement that the probability of $\alpha_0 \epsilon CI(X, 1 - \gamma)$ is at least $1 - \gamma$, i.e., that
$P_{\alpha_0}(F(X - 1, \cdot)^{-1}(1 - \gamma/2) \geq \alpha_0 \text{ or } F(X, \cdot)^{-1}(\gamma/2) \leq \alpha_0) = P_{\alpha_0}(\alpha_0 \notin CI(X, 1 - \gamma)) \leq \gamma$.

The BiasedUrn function pFNCHypergeo computes the function $F(x, a)$ through the identity:

$$F(x, \alpha) = F(x, m_A, m_B, N, \alpha) = \text{pFNCHypergeo}(x, m_A, N - m_A, m_B, e^\alpha)$$

The test-based confidence interval $CI(X, 1 - \gamma)$ defined above in (4) is precisely the interval CI.CP, labeled CP because of the close resemblance of this test-based idea to that of Clopper and Pearson [19] in a famous 1934 paper about confidence intervals for binomial proportions. By definition, CI.CP is *conservative* in the sense that its probability of covering the true $\alpha$ value

generating the random variable $X$ is at least $1 - \gamma$. However, due to the discreteness of the Extended Hypergeometric distribution, for moderate $m_A$, $m_B$, $N$ the excess of the actual coverage probability over the nominal $1 - \gamma$ can sometimes be quite large, up to 0.03 or more when $\gamma = 0.05$.

**Blaker confidence interval (CI.Blaker).** Another type of confidence interval defined by [20] is provably conservative and also a subset of CI.CP, and is found for some combinations of $(\alpha, X)$ to improve the coverage probability substantially. Defining and justifying the construction of Blaker Confidence Interval (CI.Blaker) is a little more abstract and difficult, and we refer the interested reader to the Splus code and Theorem 1 of [20], defined in that paper for unknown Binomial proportions, that we have adapted in our package functions AcceptAffin and AcceptAffCI.

The coverage performance of the CI.CP in the context of Binomial proportions, was compared by [21] with several other popular confidence intervals, mostly inspired by the large-sample closeness (the DeMoivre-Laplace Central Limit Theorem) of the binomial to the normal distribution with the same mean and variance. They found CI.CP to be extremely conservative, while several well-known and widely used large-sample-theory-based intervals exhibited (for some $n$, $p$ combinations) extremely erratic performance, even for surprisingly large values of $n$. (That observation, and some theoretical explanation of it based on deeper large-sample theory, was the main point of the [21] article and other theoretical articles on which it was based.) Although that study compared several confidence intervals, it did not study the modified Clopper-Pearson intervals known as midP (CI.midP) [[22], p. 605]. Both the Clopper-Pearson and midP intervals are applicable to small- as well as large-sample data and have natural generalizations to unknown scalar parameters for other discrete random variables like the Extended Hypergeometric.

**Preference for conservative or close-to-nominal coverage intervals?.** Should confidence intervals always be conservative to be useful? For some applications—such as the confirmatory clinical trials required by governmental statistical regulatory agencies (e.g., FDA, NIH, NIST) and certain laboratory testing—the answer is yes. But for a great deal of exploratory work in science, including ecology, it will often be preferable to choose a type of confidence interval with coverage probability reliably as close as possible to the nominal coverage $1 - \gamma$, even at the cost of occasional under-coverage by a few percent.

With this in mind, we present two alternative confidence intervals for the log-affinity parameter $\alpha$, the midQ and midP intervals. These intervals follow the same strategy as CI.CP but calculate interval endpoints differently. Instead of using distribution-function values that jump at each successive integer co-occurrence value, midQ and midP use linearly interpolated values. midQ computes endpoints from linearly interpolated inverse distribution function values between successive integers, whereas midP computes endpoints from linearly interpolated distribution function values. As it turns out, these two approaches give nearly identical results (see below).

**midQ confidence interval.** As explained in "Advances: The median interval", we define the interval of all $\alpha$ values for which the observed co-occurrence count $X$ is the $q$'th quantile for $F(x, m_A, m_B, N, \alpha)$ as:

$$(F(x - 1, \cdot)^{-1}(q), F(x, \cdot)^{-1}(q)). \tag{5}$$

To estimate a single $\alpha$ value for which $X$ is a $q$-th quantile, we choose the midpoint of the interval (5). Then a natural choice for an interval of $\alpha$ values compatible with the observed co-

occurrence count $X$, expressed in terms of the quantiles $\gamma/2$ and $1 - \gamma/2$, is the interval:

$$CI.midQ = \left( {}^{1\!/_2}\left(F(X, \cdot)^{-1}(1 - \gamma/2) + \left(F(X - 1, \cdot)^{-1}(1 - \gamma/2)\right)\right), \right.$$
$$\left. {}^{1\!/_2}\left(F(X, \cdot)^{-1}(\gamma/2) + \left(F(X - 1, \cdot)^{-1}(\gamma/2)\right)\right) \right) \tag{6}$$

This interval has coverage probability closer to the nominal level $1 - \gamma$ than the conservative intervals CI.CP and CI.Blaker described above, though at the cost of occasional undercoverage (e.g., up to 0.03 for 95% intervals, for certain combinations of $\alpha$ and $m_A$, $m_B$, $N$).

**midP confidence interval.**   Another confidence interval Midpoint Exact or MidP (CI.midP), which has almost exactly the same behavior for most $(X, m_A, m_B, N)$ as CI.midQ, is defined directly in terms of approximate distribution function values rather than quantiles. This interval, named midP by analogy with the analogous interval for unknown binomial proportions [[22], p. 605], is defined exactly like CI.CP with the distribution function $F(x, \alpha)$ linearly interpolated:

$$CI.midP = \{\alpha : {}^{1\!/_2}\left(F(X - 1, \alpha) + F(X, \alpha)\right) \in (\gamma/2, 1 - \gamma/2)\} \tag{7}$$

The intervals described so far are defined only for values of $X$ that differ from the extremes of its range. When $X = \max(m_A + m_B - N, 0)$, the left endpoint of the confidence interval should be $-\infty$, but this is replaced by $-\log(2N^2)$, which, as mentioned earlier, is a provable lower-bound for all $\hat{\alpha}$ corresponding to $X > \max(m_A + m_B - N, 0)$. Similarly, when $X = \max(m_A, m_B)$ the right endpoint of all confidence intervals is taken to be $\log(2N^2)$. One further modification in the CooccurrenceAffinity package, adopted to preserve precision of the confidence intervals computed using the BiasedUrn package evaluations of $F(x, \alpha)$, is the restriction that all confidence intervals are replaced by their intersection with the interval $(-M, M)$, where $M$ denotes the value at which $\hat{\alpha}$ is capped.

**Good practice.**   The theoretical discussion of confidence intervals for $\alpha$ presented under "Advances: Four types of confidence interval," along with the examples below, may help in deciding which CI to use. The trade-off between intervals with conservative versus near-nominal coverage is related to interval length. The conservative CI.CP and CI.Blaker 95% intervals include the population parameter at least 95% of the time (i.e., the true coverage probability $\geq 0.95$). In confirmatory applications, where strict control of the type I error probability is desirable, we recommend CI.Blaker, as it is the shortest conservative interval that we know.

In exploratory studies, where many confidence intervals for different datasets might be compared, interval coverage probabilities (i.e., probabilities of CIs covering the true $\alpha$) should ideally be as close as possible to each other and to the nominal coverage probability, usually 0.95. In such studies, we recommend either of the nearly indistinguishable midP or midQ intervals because their 95% CIs are narrower than the conservative alternatives, and the true coverage probabilities tend to center around 0.95, with approximately equal probabilities of missing the population parameter in the positive and negative directions. But when the 2x2 table is nested, and the co-occurrence count is at (or maybe very near to) one of its logical extremes, we advocate the reporting of one-sided confidence intervals.

## Overview of CooccurrenceAffinity package

Fig 3 displays the functions of this package visually, color-coded based on their role, with their relationships to each other indicated by proximity and arrow. Several of these functions rely on the output of another R package BiasedUrn.

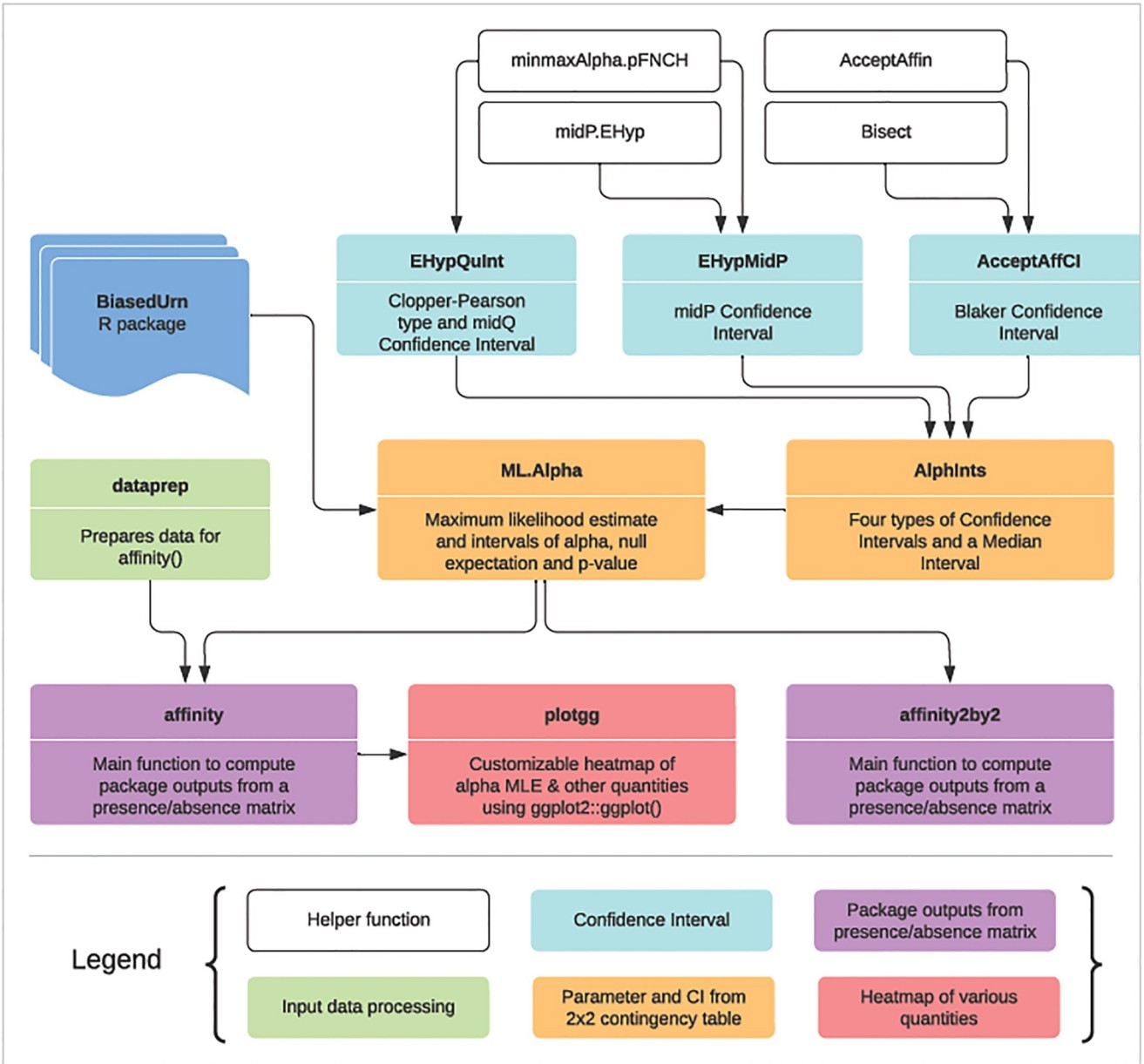

**Fig 3. Workflow of CooccurrenceAffinity.** The functions are grouped by color based on their role, as indicated in the box. A function at the base of an arrow feeds its output to the function at the tip of the arrow. Not all functions of the package are shown here.

### Functions to compute four types of confidence intervals

EHypQuInt computes Clopper-Pearson type Confidence Interval and midQ Confidence Interval. AcceptAffCI and EHypMidP compute Blaker Confidence Interval and midP Confidence Interval, respectively. These three functions rely on the following four helper functions for computation of confidence intervals: minmaxAlpha.pFNCH, AcceptAffin, Bisect, and midP. EHyp.

### Function for parameter estimate and all intervals (median and CI)

AlphInts computes and lists the four confidence intervals and the median interval. It also reports null-distribution expected co-occurrence count and p-value. ML.Alpha includes all the outputs of AlphInts and additionally computes the parameter estimates ($\hat{\alpha}$) and the maximized log-likelihood.

### Functions for analysis of 2×2 contingency table

AlphInts and ML.Alpha can be used for co-occurrence analysis based on a 2×2 contingency table show in Fig 1. However, if the goal is also to obtain the values for the common traditional indices, affinity2by2 can be useful. affinity2by2 supplements the outputs of ML.Alpha with the indices of Jaccard, Sørensen–Dice, and Simpson. These indices are useful for reference to older studies but should be used with caution.

### Function for analysis of presence-absence matrix

A presence-absence matrix (e.g., species-by-site matrix) is analyzed using affinity. The input dataset is examined for potential issues and a ready-to-analyze matrix is prepared by calling a separate function dataprep. A user should indicate whether rows or columns are being analyzed. For a species-by-site matrix, if species are given in rows and islands in columns, an analysis of rows gives affinity between every pair of species, whereas an analysis of columns gives affinity between every pair of islands. A user can also optionally select certain rows or columns based on their position or name. This function accepts abundance data as input. In such cases, a user should set datatype = "abundance" and select a threshold to convert abundance data to binary. The user should also indicate whether the threshold should be categorized as presence or absence using "class0.rule" argument.

Affinity, by default, returns two matrices. The first one is a long-format data frame with each pairwise analysis on a row and a total of 19 columns showing various outputs of analysis. The second matrix is the processed presence-absence matrix used for the analysis (e.g., binary version of the abundance matrix, a subset of the original matrix with selected rows/cols). Optionally, with "squarematrix" argument, affinity returns up to 11 additional square matrices for $\hat{\alpha}$ and other quantities. A square matrix contains the entities being analyzed both in rows and columns in the same order, with each cell of the matrix holding the values of a particular quantity (e.g., $\hat{\alpha}$).

### Function to plot the output

plotgg generates a heatmap for any quantitative column in the long-format output available under $all of affinity output. plotgg utilizes ggplot2::ggplot on the backend. The arguments of plotgg make it easy to customize the plot.

### Function to test true coverage of confidence intervals

CovrgPlot computes the true coverage probabilities of the selected confidence intervals and returns, for each confidence interval type, a multi-panel plot with (a) line plot of the coverage probabilities by $\hat{\alpha}$, and (b) a histogram of the coverage probabilities, showing percentage of the probabilities falling below (rate of failure to include population parameters) *vs*. above the confidence level. Following a plan similar to that described computationally in [21] regarding confidence intervals for Binomial proportions, our package function CovrgPlot calculates and plots the curve of coverage probabilities under the Extended Hypergeometric distribution for fixed ($X$, $m_A$, $m_B$, $N$) and *all $\alpha$* values.

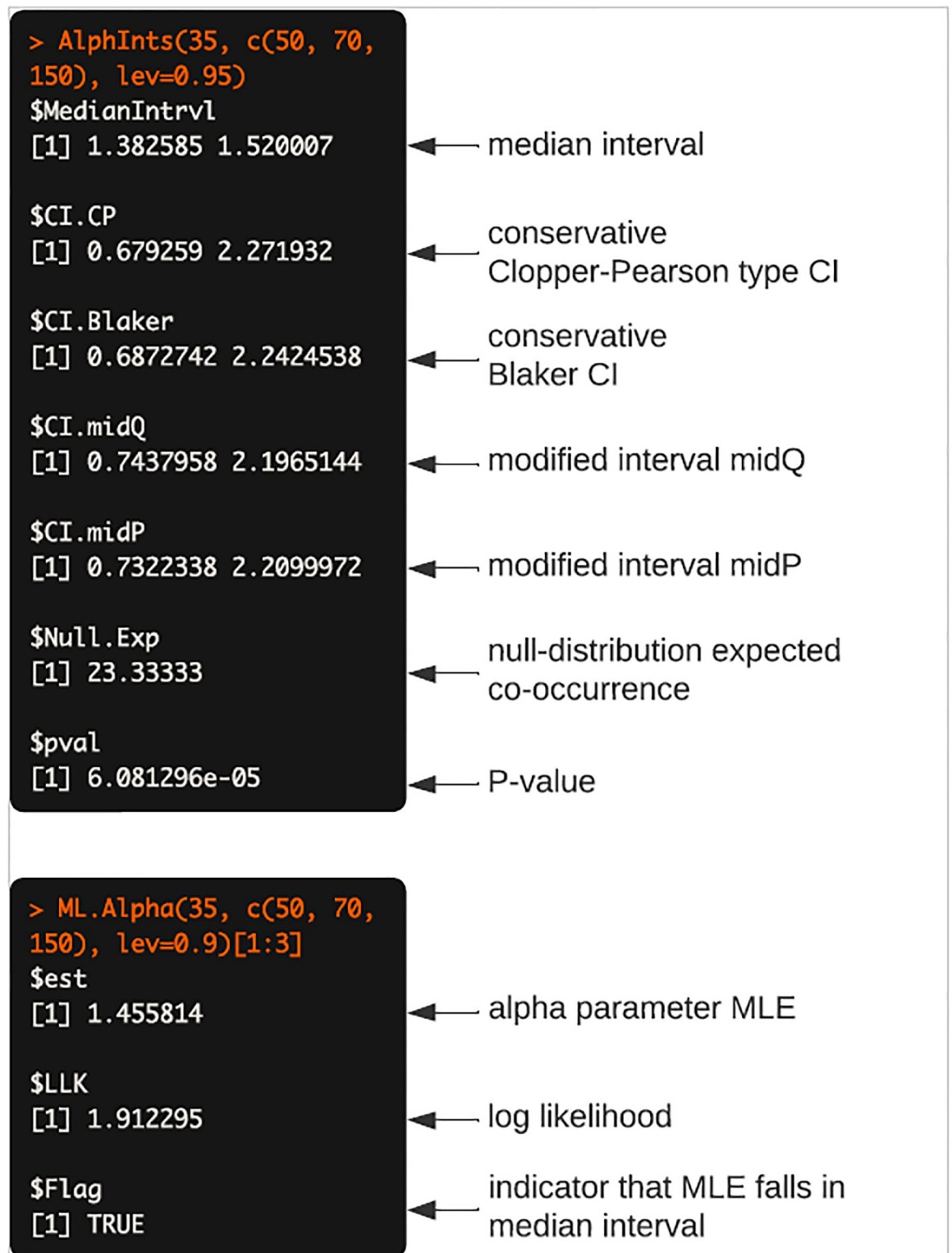

**Fig 4. An example of syntax and output of AlphInts and ML.Alpha.**

```
> par.fit <- loglin(array(c(35,15,35,65),c(2,2)), list(1:2),
+    start = rep(1,4), eps = 0.1, iter = 20, param = T)$param
2 iterations: deviation 0

> c(log(35*65/(15*35)),  4*par.fit$'1.2'[[1]])
[1] 1.466337 1.466337
```

Fig 5. The log cross-product ratio of the example in Fig 4.

Covrg is a way to generate the set of 4 coverage probability values without plotting. The coverage probabilities are calculated (and stored as an output array) in CovrgPlot at values where the coverage curve changes direction, but can be separately calculated at any single $\alpha$ using Covrg.

## Examples and illustrations

### Analysis of 2×2 contingency table

We compute with a running example: $X = 35$, $m_A = 50$, $m_B = 70$, $N = 150$. The syntax and results of the function calls for computing the MLE $\hat{\alpha}$, the median interval, and the 95% two-sided equal-tailed confidence intervals for $\alpha$, are shown in Fig 4.

The conservative interval of Clopper-Pearson type CI is definitely wider than other intervals (Blaker, midQ and midP). The MLE computed here is very close to the log cross-product ratio, which is also what the loglin base-package loglinear-modeling function gives, as in Fig 5.

As demonstrated above, the MLE $\hat{\alpha}$ in this instance is 1.455814. The difference between this estimate and the loglin output is that MLE here conditions on the marginals, while loglin does not. Another method of computing MLEs is Poisson regression, which also does not condition on marginals and produces the same answer as loglin. We recommend the MLE conditioned on marginals as making the most sense for ecological analyses.

### Analysis of presence-absence matrix

For users with presence-absence matrix interested in several pairwise analyses, affinity is particularly useful. For each pair of entities, affinity computes the 2×2 contingency table from presence-absence data and performs the analysis. An example showing computation of affinity between species or between sites from the same dataset is presented in Fig 6.

### Quantifying discreteness versus coverage

The median interval width is a measure of the discreteness of the co-occurrence distribution, while the confidence intervals retain their usual interpretation. The practical interpretation is that the CI's are imprecise (due to discreteness) roughly by the magnitude of median-interval width.

We illustrate an example of the relative sizes of the median interval and confidence interval and their positioning with respect to the MLE (Fig 7). Examining the trend for all possible values of $X$ from 1 to 49, we observe that the estimated median interval and Blaker CI tend to

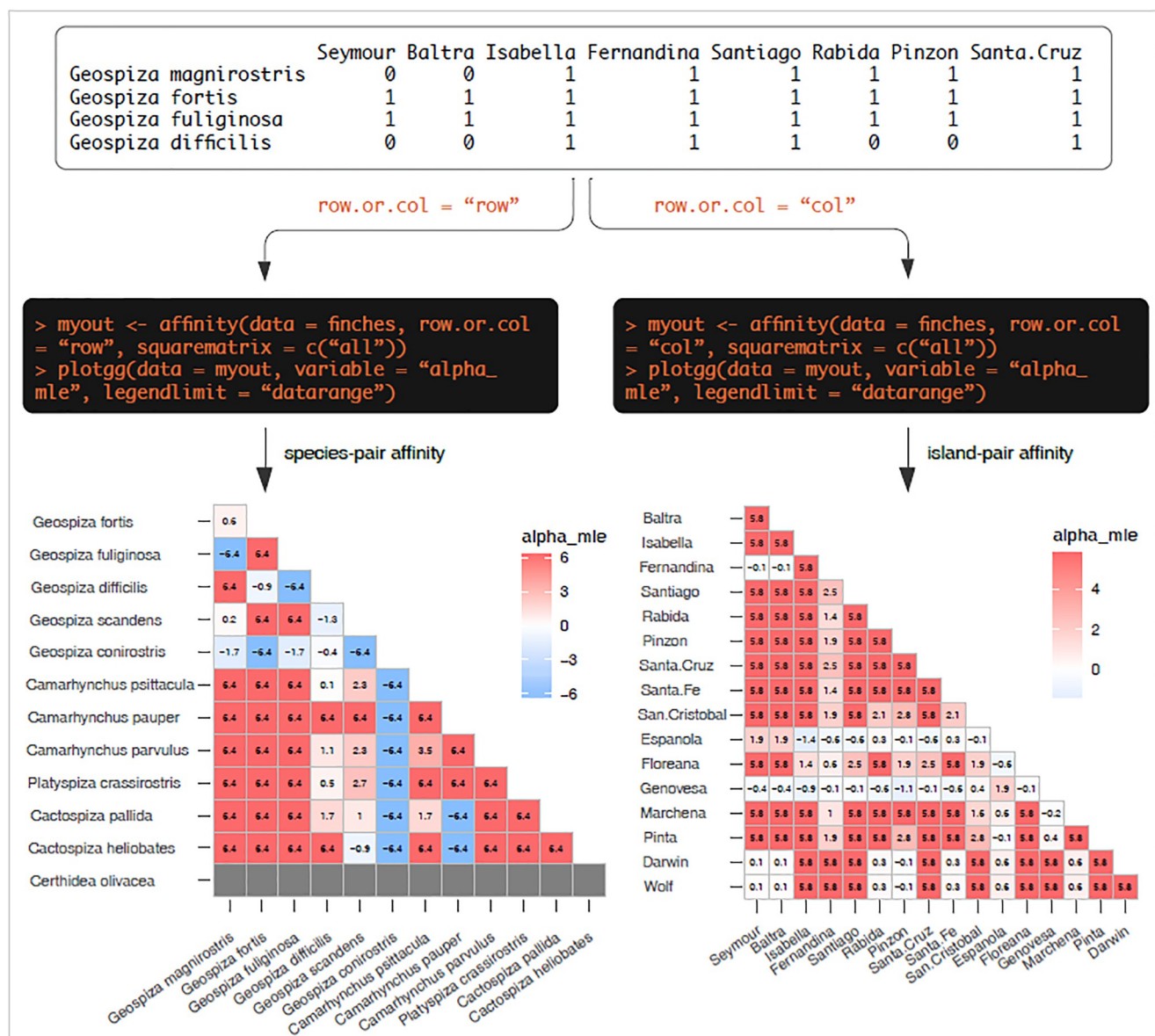

**Fig 6. Analysis of a presence-absence matrix for computing several pairwise analyses for row or column.** The top panel shows a portion of the dataset. The code block shows how easily affinity between species (rows) or between islands (columns) can be computed using affinity, and how the results can be visualized using plotgg.

become wider for more extreme co-occurrence values of $X$, while the median interval is always much shorter than the Blaker CI (Fig 7a, other CIs not shown). Blaker CI and Clopper-Pearson type CI tend to be very similar in length and positioning when the co-occurrence count is near its null expectation, while the Blaker CI tends to be shorter than the Clopper-Pearson type CI for $X$ values farther from that null expectation, which is 50*70/150 = 23.3 in the Fig 7b.

Fig 8 further clarifies the construction of the confidence intervals, showing function values $F(x, \alpha)$ for various choices of $x$. Note that the values $F(x, \alpha)$ decrease as a function of $\alpha$, because larger $\alpha$ corresponds to larger co-occurrence random variable $X$ and smaller probability for that $X$ to be $\leq x$. As an example of the information to be read from the figure, for X = 20: the

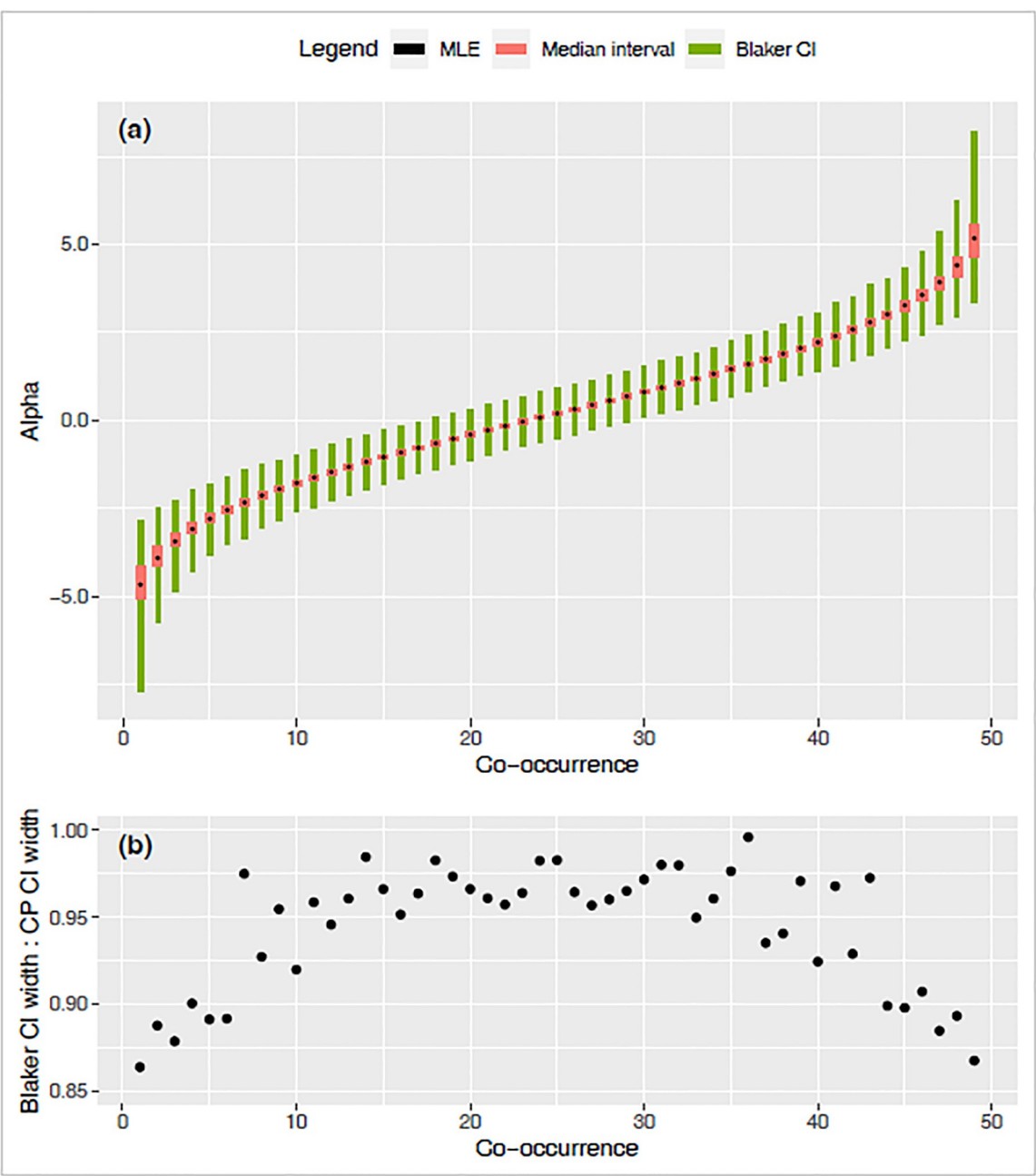

**Fig 7. Intervals and MLE for $\alpha$ for all possible $X$ values (co-occurrence) in an example of $m_A = 50$, $m_B = 70$, $N = 150$.** (a) MLE, median interval and Blaker CI (95%), (b) ratio of the length of Blaker CI (95%) to that of Clopper-Pearson type CI (95%).

median-interval in green horizontal bar is the alpha-interval between the pair of green curves at height $F(X) = 0.5$, with $\hat{\alpha}$ the alpha-coordinate of the solid black circle between the two curves. The Clopper-Pearson confidence interval has left endpoint where the left green curve crosses level 0.975 and right endpoint where the right green curve crosses level 0.025.

The four types of CIs differ, which has implications for their true coverage probability. By inspecting the trend of probabilities, we can see several patterns (Fig 9). First, the true coverage

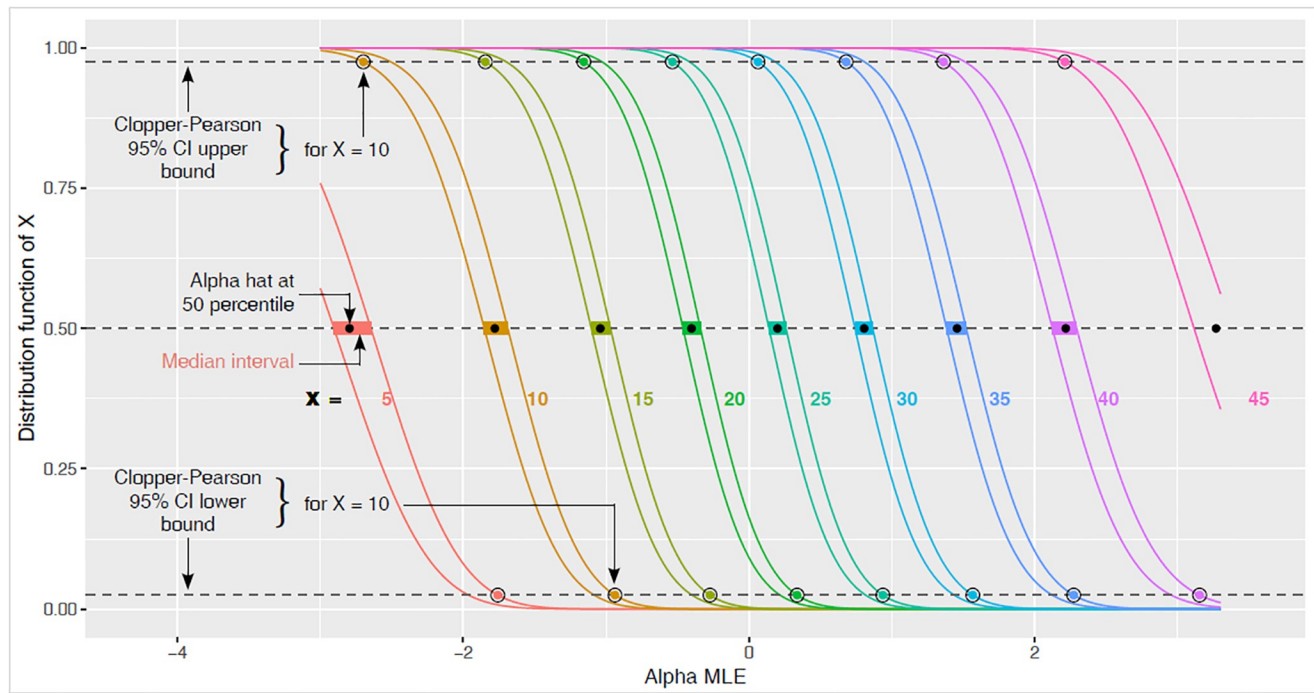

**Fig 8. Functions $F(x, \alpha)$ plotted for selected pairs $x$, $x − 1$ and all $\alpha$ in the range $(−3, 3.3)$ for the same example of $m_A = 50$, $m_B = 70$, $N = 150$.** The colored sigmoid lines are the curves $F(X, \cdot)$, and immediately to the left of each such curve is another (with the same color) for $X − 1$. Horizontal dashed lines are the quantile levels 0.025, 0.50, 0.975. Black solid circles are plotted for the $X$ values in each curve-pair at $(\hat{\alpha}, 0.5)$. Colored solid circles encircled by black circles are plotted for the upper and lower bounds of Clopper-Pearson 95% confidence intervals. The median-interval of $\alpha$'s for each $X$ is the colored horizontal bar connecting the curve-pair at height 0.5.

probability for 95% Clopper-Pearson type CI (Fig 9a and 9b) and Blaker CI (Fig 9c and 9d) is always 95% or more (histograms show 100% of the mass above the 0.95 probability). In order to minimize the probability of not including the population parameter, these two CIs become too wide. However, the Clopper-Pearson CI interval is (unnecessarily) wider than the Blaker CI interval; the latter is closer (within 0.025) to the correct coverage compared to the former (within 0.05). Second, the 95% midP (Fig 9e and 9f) and midQ (Fig 9g and 9h) interval are narrower than the other two; the true coverage probabilities of midQ and midP tend to center around 0.95 for the range of $\alpha$ values seen with all possible co-occurrences. Roughly half the time (53.59% for midQ and 45.3% for midP, shown in histograms), these CIs have actual coverage probability below (but within 0.02 of) the nominal value 0.95.

The coverage of the midP type interval is close to the correct coverage (within ±0.02) and averages across $\alpha$ to almost exactly the nominal 95%, while the Clopper-Pearson type interval is conservative by as much as 0.05. This picture relates to only one $(m_A, m_B, N)$ combination, but is the result of a rapid exact calculation, not a simulation, and so can easily be reproduced for other fixed margin values.

## Conclusion

Cooccurrence Affinity is a new R package that computes parameters associated with a recently developed metric of association in co-occurrence data [11]. This novel metric, called alpha hat ($\hat{\alpha}$), corrects for the pervasive errors in analysis of co-occurrence data with traditional indices. This package includes flexible functions for (1) analyzing 2×2 contingency tables of counts of

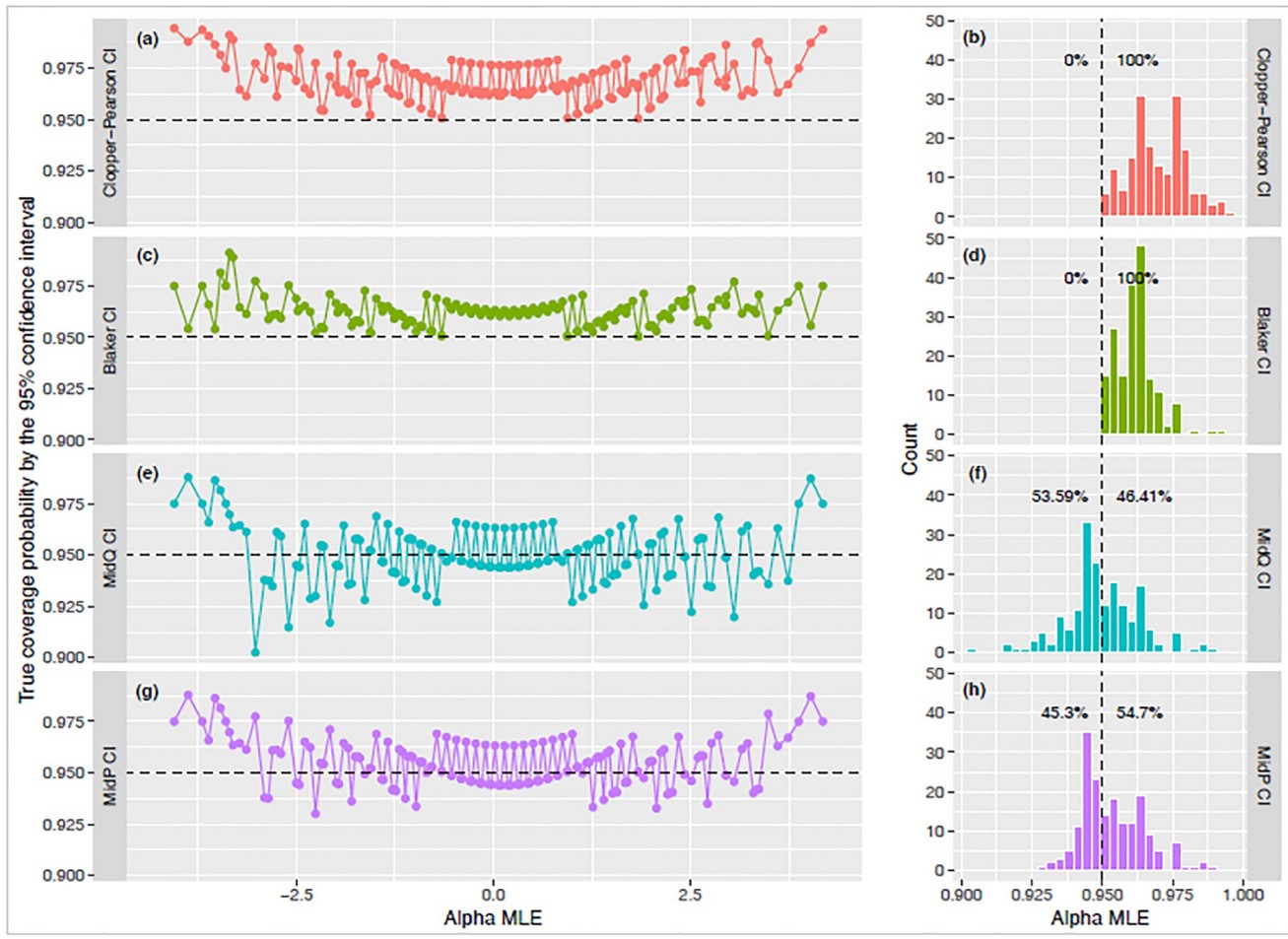

**Fig 9. True coverage probability of the four types of confidence intervals for the example presented in** Fig 7 **with $\alpha$ values shown from -4 to 4, computed as CovrgPlot(c(50,70,150), lev = 0.95).** The true coverage probability of the 95% Clopper-Pearson (a) or Blaker (c) CI's are always at least 95% (dashed line) for all co-occurrences resulting in a histogram with its entire mass above 0.95 (b,d). Plots of true coverage probability are also shown for midQ (e,f) and midP (g,h) CIs.

occurrences or the more widely used format of m×n matrix of presence-absence, and (2) plotting the analysis output. The novel elements of this paper are the introduction of median interval and four types of confidence intervals of alpha. The package provides functions to (a) compute these intervals, and (b) assess their true coverage probability of including the population parameter, and (c) visually and numerically evaluate the trade-offs between interval length and the probability failing to include the population parameter $\alpha$.

We anticipate that this package will serve as a user-friendly tool for estimating affinity in co-occurrence data. Given the increasing integration of environmental and spatial datasets into cloud computing environments—where Python is becoming the preferred choice for data science—a Python adaptation of this package would greatly benefit users.

## Author Contributions

**Conceptualization:** Kumar P. Mainali.

**Formal analysis:** Kumar P. Mainali, Eric Slud.

**Funding acquisition:** Kumar P. Mainali.

**Methodology:** Eric Slud.

**Software:** Kumar P. Mainali, Eric Slud.

**Visualization:** Kumar P. Mainali.

**Writing – original draft:** Kumar P. Mainali, Eric Slud.

**Writing – review & editing:** Kumar P. Mainali, Eric Slud.

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
