## [Decision Letter · Decision Letter 0]

4 Oct 2024

PONE-D-24-38304CooccurrenceAffinity: An R package for computing a novel metric of affinity in co-occurrence data that corrects for pervasive errors in traditional indicesPLOS ONE

Dear Dr. Mainali,

Thank you for submitting your manuscript to PLOS ONE. After careful consideration, we feel that it has merit but does not fully meet PLOS ONE’s publication criteria as it currently stands. Therefore, we invite you to submit a revised version of the manuscript that addresses the points raised during the review process.

We look forward to receiving your revised manuscript.

Kind regards,

Abhik Ghosh

Academic Editor

PLOS ONE

Journal Requirements:

4. Thank you for stating the following financial disclosure: “KM was supported by the Grayce B. Kerr Fund, Inc, and by the National Science Foundation DBI-1639145 under funding received for the National Socio-Environmental Synthesis Center (SESYNC).”

5. Please provide a complete Data Availability Statement in the submission form, ensuring you include all necessary access information or a reason for why you are unable to make your data freely accessible. If your research concerns only data provided within your submission, please write "All data are in the manuscript and/or supporting information files" as your Data Availability Statement.

6. Please amend the manuscript submission data (via Edit Submission) to include author Eric Slud.

Reviewers' comments:

Reviewer's Responses to Questions

**Comments to the Author**

1. Is the manuscript technically sound, and do the data support the conclusions?

Reviewer #1: Partly

Reviewer #2: Yes

2. Has the statistical analysis been performed appropriately and rigorously? 

Reviewer #1: Yes

Reviewer #2: Yes

3. Have the authors made all data underlying the findings in their manuscript fully available?

Reviewer #1: Yes

Reviewer #2: Yes

4. Is the manuscript presented in an intelligible fashion and written in standard English?

Reviewer #1: No

Reviewer #2: Yes

5. Review Comments to the Author

Reviewer #1: Manuscript PONE-D-24-38304 titled “CooccurrenceAffinity: An R package for computing a novel metric of affinity in co- occurrence data that corrects for pervasive errors in traditional indices” presents a new R package – CooccurrenceAffinity. This package can be used to analyze species co-occurrence data an represent them using graphical visualization. To this end, the authors use a new metric – alpha, assuming null distribution or randomization of the association, thus adding to the accuracy of the analysis.

Here are my thoughts.

Try and avoid repetitions of phrases between title and keywords.

Line 18 – please elaborate the phrase “many problems” in about a line. You may wish to combine part of the trailing line as well.

Minor grammatical errors relating to verb and article use are present in the manuscript. For example, line 30 among others.

The major concern with this manuscript is that the authors have chosen to use a very confusing approach towards their main goal. The index as proposed by the authors and claimed to be free from errors unlike traditional methods.

The proposed affinity index relies on assumptions similar to the fixed row-equiprobable column (FE) null model, which is not mentioned. This makes the claims of the authors regarding novelty as overestimated, as the method is not as original as is led to believe.

The second critique in my opinion is the assumption that all sites are equally suitable for species, which may not hold in real-world datasets with heterogeneous sites, such as islands or even populations dispersed along gradients. The proposed method may underperform in such cases. The assumption of equiprobability needs to be revisited and established with proper justification i.e., limitations of the presented theory.

My third concern about the R package's implementation is inconsistent results from the same data. The reliability of the package is brought into question. This limits practical applicability.

Reviewer #2: The manuscript titled "CooccurrenceAffinity: An R package for computing a novel metric of affinity in co-occurrence data" introduces a valuable tool for ecological and biogeographical research. The CooccurrenceAffinity R package offers a new affinity metric, α, addressing the limitations of traditional co-occurrence indices like Jaccard and Sørensen-Dice. The package's ability to compute maximum likelihood estimates (MLE) of α, along with four different confidence intervals, makes it a powerful tool for researchers dealing with binary presence-absence data. The manuscript is well-structured and provides detailed explanations of the methods and the theoretical underpinnings of the metric. The integration of this package into ecological workflows can provide more reliable interpretations of species associations, which is a significant contribution to the field. However, some areas need clarification and improvement, both in terms of mathematical rigor and writing style, to enhance the paper's clarity and utility for readers.

Comment1:

The manuscript heavily relies on MLE for α, which can lead to biased results in small sample sizes, a common issue in ecological studies. It would be beneficial to include a discussion on the limitations of MLE in such contexts. Consider exploring alternative methods, such as bias-corrected MLE or Bayesian estimators, which may offer more robust results when working with small datasets. Adding such a discussion would improve the paper's applicability across a wider range of sample sizes.

Comment2:

The use of hat α values to avoid ±∞ for extreme co-occurrence values introduces potential distortions in the results. It's important to further explain the impact of these hat values on interpretation and provide clear guidance for users on how to handle such cases in biological applications.

Comment3:

While the manuscript presents four types of confidence intervals, it does not fully explain the trade-offs between more conservative intervals (e.g., Clopper-Pearson) and exploratory ones (e.g., midP, midQ). Including a more detailed discussion on when to use each type of CI, especially in exploratory versus confirmatory research, would enhance the utility of the method. Guidance on managing undercoverage risks in exploratory studies would also be helpful.

Comment4:

The paper overlooks the need for adjusting for multiple comparisons when analyzing numerous species pairs, which can increase the risk of Type I errors. Addressing this by suggesting corrections such as Bonferroni or False Discovery Rate (FDR) would enhance the reliability of the results in large-scale ecological analyses, where multiple pairwise comparisons are common.

Comment5:

The assumption of independent co-occurrences ignores spatial autocorrelation, which is crucial in ecological studies. Acknowledging this limitation and suggesting future extensions to incorporate spatial dependence would make the method more broadly applicable to real-world scenarios where spatial structure plays a significant role.

Comment6:

The ecological interpretation of α as a log-odds ratio may not be intuitive for all readers. Providing additional examples or practical explanations of what different values of α represent in terms of species interactions (e.g., competition, facilitation) would make the metric more accessible and meaningful for ecologists.

Comment7:

Some sentences in the manuscript are overly long and complex, making them difficult to follow. Simplifying these sentences and using clearer, more concise language will improve the readability and overall flow of the manuscript.

Minor

Comment1:

The manuscript uses passive voice excessively, which can make the writing less engaging. For instance, “It was found that the MLE provided reliable estimates” could be improved by switching to active voice: “We found that the MLE provided reliable estimates.” This would make the writing clearer and more direct.

Comment2:

Some terms are not clearly defined when first introduced. For example, “Affinity is calculated using the log-odds ratio” is unclear without defining what affinity or the log-odds ratio means. A clearer explanation would help, such as: “Affinity refers to the degree of association between two species based on their co-occurrence. The log-odds ratio quantifies this relationship, indicating how much more likely species are to co-occur than by chance.”

Comment3:

Some sentences are too long and include multiple ideas, making them difficult to follow. For example: “The novel metric of affinity, α, is introduced to resolve issues in traditional indices, and it is computed through the MLE, which provides a better estimation for species association, making it more reliable for large datasets, although small datasets may still present challenges.” This could be broken down into clearer sentences: “The novel metric of affinity, α, resolves issues in traditional indices. It is computed using the MLE, providing a more reliable estimation for species association in large datasets. However, small datasets may still present challenges.”

Comment4:

The abstract is wordy and lacks a clear focus on the paper’s key contributions. For example, “We provide functions for analysis and plotting based on various data formats, and compute the novel metric along with traditional indices such as Jaccard and Simpson.” This could be made more concise, such as: “We introduce the CooccurrenceAffinity R package, which computes a novel metric of species affinity (α) and corrects biases in traditional indices. The package provides functions for analyzing co-occurrence data and generating visual outputs.”

6. PLOS authors have the option to publish the peer review history of their article (what does this mean?). If published, this will include your full peer review and any attached files.

Reviewer #1: No

Reviewer #2: No

---

## [Author Response · Author response to Decision Letter 0]

20 Nov 2024

We have provided a detailed document as a separate upload responding to the reviewers' comments.

---

## [Decision Letter · Decision Letter 1]

16 Dec 2024

CooccurrenceAffinity: An R package for computing a novel metric of affinity in co-occurrence data that corrects for pervasive errors in traditional indices

PONE-D-24-38304R1

Dear Dr. Mainali,

We’re pleased to inform you that your manuscript has been judged scientifically suitable for publication and will be formally accepted for publication once it meets all outstanding technical requirements.

Kind regards,

Abhik Ghosh

Academic Editor

PLOS ONE

Additional Editor Comments (optional):

Reviewers' comments:

Reviewer's Responses to Questions

**Comments to the Author**

1. If the authors have adequately addressed your comments raised in a previous round of review and you feel that this manuscript is now acceptable for publication, you may indicate that here to bypass the “Comments to the Author” section, enter your conflict of interest statement in the “Confidential to Editor” section, and submit your "Accept" recommendation.

Reviewer #2: All comments have been addressed

2. Is the manuscript technically sound, and do the data support the conclusions?

Reviewer #2: Yes

3. Has the statistical analysis been performed appropriately and rigorously? 

Reviewer #2: Yes

4. Have the authors made all data underlying the findings in their manuscript fully available?

Reviewer #2: Yes

5. Is the manuscript presented in an intelligible fashion and written in standard English?

Reviewer #2: Yes

6. Review Comments to the Author

Reviewer #2: (No Response)

7. PLOS authors have the option to publish the peer review history of their article (what does this mean?). If published, this will include your full peer review and any attached files.

Reviewer #2: No

---

## [Editor Report · Acceptance letter]

26 Dec 2024

PONE-D-24-38304R1 

PLOS ONE

Dear Dr. Mainali, 

I'm pleased to inform you that your manuscript has been deemed suitable for publication in PLOS ONE. Congratulations! Your manuscript is now being handed over to our production team.

Kind regards, 

on behalf of

Dr. Abhik Ghosh 

Academic Editor

PLOS ONE